# Boosting Language Model Fine-Tuning via Zeroth-Order Hybrid Methods with Additional Memory Aid

## Abstract

When adjusting large language models (LLM) for downstream applications, parameter-efficient fine-tuning (PEFT) significantly reduces memory costs. However, due to the need to store the activation values of backpropagation during gradient computation, traditional First-order (FO) fine-tuning algorithms generate a large amount of memory overhead. Zeroth-order (ZO) algorithms eliminate the need for activation storage by approximating gradients using finite differences of function values, providing a feasible solution when GPU memory is insufficient. However, the existing ZO methods have the problem of slow convergence, and they have far from realized the potential memory advantage of dual forward propagation. In this paper, a low-rank ZO gradient estimation method is proposed, which uses low-rank fast calculation and stable sampling strategy to accelerate the convergence of the model. Simultaneously, we divide the model into different hierarchical blocks, optimize the shallow blocks using the low-rank ZO optimizer, and perform FO optimization on the deepest blocks (closest to the output) to accelerate convergence. We further propose memory offloading scheduling, offloading the hierarchical blocks that have already participated in computation into CPU memory, and only moving the blocks that need to be calculated into GPU memory. By using this method, we can fine-tune very large models, such as the OPT-175B with over 175 billion parameters, on a GPU with only 17GB memory, while maintaining a relatively fast convergence speed and fine-tuning performance close to the FO algorithm.

## 1 Introduction

Large language models (LLMs) have exhibited remarkable capabilities in diverse fieldsAchiam et al. (2023); Brown et al. (2020); Solaiman et al. (2019). Fine-tuning pre-trained models has emerged as the standard strategy for tailoring LLMs to particular downstream tasks Gururangan et al. (2020); Sanh et al. (2021). Parameter-efficient fine-tuning (PEFT) techniques, like those introduced by Hu et al. (2021b); Lester et al. (2021), seek to minimize memory usage by keeping the majority of pre-trained weights fixed and adjusting only a limited number of parameters. Nonetheless, even employing these methods, first-order (FO) optimization algorithms such as stochastic gradient descent (SGD) Amari (1993) and Adam Kingma & Ba (2014) continue to face significant memory burdens because of the requirement to retain activation values for back-propagation in gradient calculations. This challenge intensifies further in scenarios involving extended contexts, where activations constitute the primary source of memory consumption.

To improve memory efficiency, a promising option is to use zeroth-order (ZO) algorithms Spall (1992). Unlike first-order (FO) methods, ZO algorithms avoid explicit gradient computations; instead, they estimate gradients via finite differences of function evaluations. This removes the need for backpropagation and storing activations, yielding substantial memory savings. Over the past decades, ZO methods have been studied extensively Duchi et al. (2013); Nesterov & Spokoiny (2015); Berahas et al. (2019) and have recently been applied to fine-tuning LLMs Malladi et al. (2023b). In particular, Malladi et al. (2023b) adapts the classical ZO stochastic gradient descent (ZO-SGD) algorithm Ghadimi & Lan (2013) into a memory-optimized variant (MeZO), which reduces memory usage to about one quarter of that of standard SGD.

Nevertheless, ZO algorithms face several challenges for LLM fine-tuning. A key issue is the rank mismatch between ZO gradient estimates and true FO gradients: FO gradients obtained via backpropagation during LLM fine-tuning are typically low-rank Malladi et al. (2023b); Zhao et al. (2024b); Hao et al. (2024), whereas the ZO gradients in MeZO arise from Gaussian perturbations and are therefore nearly full-rank. Introducing a low-rank parameterization into ZO could reduce computation and potentially accelerate convergence. Despite its memory advantages, MeZO can degrade accuracy and require more optimization steps than full first-order fine-tuning across multiple tasks Malladi et al. (2023b). Moreover, because ZO avoids backpropagation, it naturally supports offloading and reloading modules between CPU and GPU without storing activations or communicating gradients, offering additional opportunities to reduce runtime GPU memory.

To focus on solving the two major issues of slow convergence speed and further memory optimization, we proposed HZO (Hybrid Zeroth-Order Optimization). The key to addressing these two challenges is to first divide the model into different hierarchical blocks, apply low-rank adaptation ZO to some shallow blocks (those closer to the input), accelerate computation. Using FO for deep blocks (those closer to the output), which avoids unnecessary backpropagation. By processing different hierarchical blocks with different methods, we can accelerate convergence. For the memory optimization issue under the HZO framework, we designed a unique scheduling algorithm that allows the hierarchical blocks involved in computation to enter GPU memory, while those that have been computed are removed from GPU memory. Compared to the MeZO Malladi et al. (2023b) algorithm, this approach has almost no additional communication time and makes it possible to fine-tune OPT-175B on a single 24G RTX4090.

In brief, the main contributions are as follows:

- We propose a low-order ZO gradient estimator. Our method uses a low-rank perturbation matrix to derive the ZO gradient, and introduce the stable sampling strategy. At the same time, using a random seed to store the matrix further reduces GPU memory consumption.

- Integrating zeroth-order and first-order optimizers for fine-tuning language models. After dividing the model into different hierarchical blocks, HZO is equipped with inter-layer mixed optimization, using the FO algorithm only for deep blocks to accelerate convergence speed.

- Compared to first-order optimizers, ZO is more suitable for CPU offloading due to the lack of communication with GPU memory for activation values and gradients. In the context of ZO optimization methods, a similar pipeline scheduling approach is applied to significantly reduce GPU memory requirements.

## 2 RELATED WORK

### 2.1 ZEROTH-ORDER OPTIMIZATION AND MEMORY-EFFICIENT FINE-TUNING

Zeroth-order (ZO) optimization estimates gradients through function evaluations, bypassing explicit backpropagation and offering a memory-efficient alternative for large model adaptation. Classical ZO methods, such as ZO-SGD, ZO-Adam, and ZO-SVRG Ghadimi & Lan (2013); Chen et al. (2019); Ji et al. (2019), often face high variance and slow convergence in high-dimensional settings. To mitigate these issues, strategies like sparse gradient estimation Balasubramanian & Ghadimi (2018) and feature reuse Chen et al. (2023) have been proposed. In parallel, memory-efficient fine-tuning techniques for large language models (LLMs), such as LoRA Hu et al. (2021a) and gradient compression Zhao et al. (2024a), significantly reduce trainable parameters or optimizer state sizes. Unlike first-order methods, ZO eliminates the need to store activations, making it appealing for LLM fine-tuning Malladi et al. (2023a). Recent work has further improved ZO efficiency via variance reduction Gautam et al. (2024), sparse masking Liu et al. (2024).

### 2.2 HYBRID OPTIMIZATION

Hybrid optimization remains relatively underexplored. Landro, Gallo, and La Grassa Landro et al. (2020) integrate SGD and Adam through the application of fixed weights to equilibrate the inputs from gradient estimations provided by each optimizer. Comparable to traditional fine-tuning

employing FO optimizers, this technique necessitates considerable memory. Ansaripour et al. Ansaripour et al. (2022) suggest hybrid optimization at the model scale within a decentralized optimization framework in a distributed setup (with certain agents refined via ZO and others via FO), which differs substantially from our approach. The hybrid optimization detailed in our research functions at the layer level (i.e., inter-layer strategy) . Some people have proposed a hybrid optimization strategy, but it consumes a lot of memory Chen et al. (2025).

## 3 METHODOLOGY

In this section, we will introduce the technical details of the proposed Hybrid Zeroth-Order Optimization (HZO), as shown in Figure 1. The left side of Figure 1 describes the hybrid optimization, and the right side describes the memory scheduling algorithm. To ensure thoroughness, we first outline the traditional ZO, as well as the MeZO Malladi et al. (2023b) method, and then delve into the technical details of our approach.

### 3.1 ZEROTH-ORDER (ZO) OPTIMIZATION

We consider the following optimization problem:

$$\min_{\boldsymbol{X}} f(\boldsymbol{X}) := \mathbb{E}_\xi[F(\boldsymbol{X}; \xi)], \tag{1}$$

where $\mathbf{X}$ denotes the set of trainable parameters with dimension $d$. For instance, in the large language model (LLM) fine-tuning process, we can express $\mathbf{X} = \{X_\ell\}_{\ell=1}^{\mathcal{L}}$ where $X_\ell \in \mathbb{R}^{m_\ell \times n_\ell}$ represents the $\ell$-th weight matrix and $\mathcal{L}$ is the total number of layers. The function $F(\mathbf{X}; \xi)$ is the loss function that depends on a random variable $\xi$.

The ZO approach approximates gradients exclusively via function assessments, bypassing the necessity for explicit gradient details. Two prevalent schemes for gradient estimation encompass the deterministic Coordinate-wise Gradient Estimation (CGE)Lian et al. (2016); Chen et al. (2023) and the Randomized vector-wise Gradient Estimation (RGE) Nesterov & Spokoiny (2015). These are formally articulated as:

$$\hat{\nabla} F(\boldsymbol{X}; \xi) := \sum_{i=1}^{d} \frac{F(\boldsymbol{X} + \epsilon \boldsymbol{E}_i; \xi) - F(\boldsymbol{X} - \epsilon \boldsymbol{E}_i; \xi)}{2\epsilon} \boldsymbol{E}_i \tag{2}$$

$$\hat{\nabla} F(\boldsymbol{X}; \xi) := \frac{F(\boldsymbol{X} + \epsilon \boldsymbol{Z}; \xi) - F(\boldsymbol{X} - \epsilon \boldsymbol{Z}; \xi)}{2\epsilon} \boldsymbol{Z}. \tag{3}$$

The scalar $\epsilon$ denotes the perturbation magnitude, which affects the accuracy of the gradient approximation. Both $E_i$ and $Z$ have the same dimensions as $X$. The quantity $E_i$ is a basis vector/matrix with its $i$-th element set to one and all other elements set to zero, whereas the elements of $Z$ are randomly generated, typically sampled independently from a standard normal distribution. An extension of the RGE method is the $q$-RGE approach. In this case, the RGE is computed $q$ times independently, and the final gradient estimate is obtained by averaging these estimations.

$$\boldsymbol{X}^{t+1} = \boldsymbol{X}^t - \alpha \hat{\nabla} F\left(\boldsymbol{X}^t; \xi^t\right) \tag{4}$$

where $\alpha$ denotes the step size, also referred to as the learning rate, and $\nabla F$ represents the gradient estimated using ZO information.

### 3.2 MEMORY-EFFICIENT ZO-SGD (MEZO)

The conventional ZO-SGD implementation involves significant memory expenses. For instance, during the creation of the gradient estimator via the RGE scheme, the classic ZO-SGD approach demands memory allocation for the perturbation matrix $Z$. To address this memory burden, the MeZO method Malladi et al. (2023b) emerged as an optimized, memory-saving alternative to ZO-SGD. In contrast to the traditional method, MeZO eliminates the need to store the full perturbation matrix $Z$. Rather, it conducts the perturbation and ZO-SGD updates directly in situ and utilizes a strategy of preserving the random seed for generating $Z$, enabling its recreation as required. Although this results in extra computational demands, it substantially lowers memory consumption.

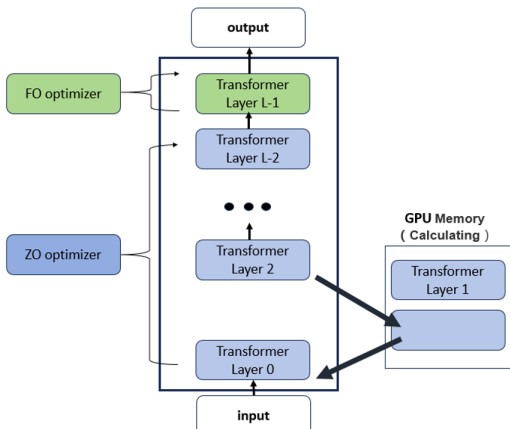

Figure 1: The middle solid line part belongs to the part temporarily stored in the CPU memory, and the right side is the block swapped into the GPU for computation.

### 3.3 LOW-RANK ZEROTH-ORDER GRADIENT ESTIMATOR

Gradients in large language models (LLMs) display low-rank characteristics. Prior research has thoroughly examined the low-rank properties inherent in neural networks Li et al. (2018); Larsen et al. (2021) . These investigations indicate that loss landscapes are confined to an intrinsic dimensionality, suggesting that model parameters can be refined effectively within a low-rank subspace. Moreover, further studies Sagun et al. (2017); Gur-Ari et al. (2018) have shown that stochastic gradients tend to converge dynamically into an exceptionally compact subspace, particularly during the fine-tuning of LLMs Zhang et al. (2023). Contemporary research Zhao et al. (2024a) further supplies theoretical justification that the gradient matrix assumes a low-rank form throughout LLM training and adaptation.

We propose a matrix-wise ZO gradient estimator, LGE, that preserves the low-rank structure in gradients. In LLM fine-tuning, let $\mathbf{X} = \{X_\ell\}_{\ell=1}^{\mathcal{L}}$ represent the model's weights, where $X_\ell \in \mathbb{R}^{m_\ell \times n_\ell}$ is the weight matrix of the $\ell$-th layer. We sample two matrices, $U_\ell \in \mathbb{R}^{m_\ell \times r_\ell}$ and $V_\ell \in \mathbb{R}^{n_\ell \times r_\ell}$, from standard normal distributions, where $r_\ell \ll \min\{m_\ell, n_\ell\}$. The LGE for the partial gradient of the $\ell$-th weight matrix is defined as

$$\hat{\nabla}_{X_\ell} F(\mathbf{X}; \xi) := \frac{F\left(\{X_\ell + \epsilon U_\ell V_\ell^T\}_{\ell=1}^{\mathcal{L}}; \xi\right) - F\left(\{X_\ell - \epsilon U_\ell V_\ell^T\}_{\ell=1}^{\mathcal{L}}; \xi\right)}{2\epsilon} \left(U_\ell V_\ell^T / r_\ell\right). \quad (5)$$

The scaling factor $1/r_\ell$ is introduced to ensure that LGE is an unbiased estimator of the true gradient as $\epsilon \to 0$ . Defining

$$U := \{U_\ell\}_{\ell=1}^{\mathcal{L}}, \quad V := \{V_\ell\}_{\ell=1}^{\mathcal{L}}, \quad r := \{r_\ell\}_{\ell=1}^{\mathcal{L}}, \quad \nabla F(X; \xi) := \{\nabla_{X_\ell} F(X; \xi)\}_{\ell=1}^{\mathcal{L}},$$

we express

$$X \pm \epsilon UV^T := \{X_\ell \pm \epsilon U_\ell V_\ell^T\}_{\ell=1}^{\mathcal{L}}, \quad UV^T / r := \{U_\ell V_\ell^T / r_\ell\}_{\ell=1}^{\mathcal{L}}.$$

Using these notations, LGE can be written into a more compact form:

$$\hat{\nabla} F(\mathbf{X}; \xi) := \frac{F\left(\mathbf{X} + \epsilon \mathbf{U}\mathbf{V}^T; \xi\right) - F\left(\mathbf{X} - \epsilon \mathbf{U}\mathbf{V}^T; \xi\right)}{2\epsilon} \left(\mathbf{U}\mathbf{V}^T / \mathbf{r}\right). \quad (6)$$

Using the definition in (5), we observe that the gradient matrix $\nabla_{X_\ell} F(X; \xi)$ has a rank of at most $r_\ell$, effectively capturing the low-rank structure of the FO gradient in LLM fine-tuning.

Following the LGE definition (6), we introduce the LGE operator

$$\text{LGE}(\mathbf{X}, \mathbf{U}, \mathbf{V}, \mathbf{r}, \epsilon, \xi) := \frac{F\left(\mathbf{X} + \epsilon \mathbf{U}\mathbf{V}^T; \xi\right) - F\left(\mathbf{X} - \epsilon \mathbf{U}\mathbf{V}^T; \xi\right)}{2\epsilon} \left(\mathbf{U}\mathbf{V}^T / \mathbf{r}\right) \quad (7)$$

To solve problem (1), the vanilla recursion with the LGE scheme is as follows. For any $t \geq 0$,

$$\boldsymbol{X}^{t+1} = \boldsymbol{X}^t - \alpha \hat{\nabla} F\left(\boldsymbol{X}^t; \xi^t\right)$$

$$\text{where} \quad \hat{\nabla} F\left(\boldsymbol{X}^t; \xi^t\right) = \text{LGE}\left(\boldsymbol{X}^t, \boldsymbol{U}^t, \boldsymbol{V}^t, \boldsymbol{r}, \epsilon, \xi^t\right). \tag{8}$$

In practice, we only need to store $U_\ell$ and $V_\ell$ for each layer $\ell$, and we apply the perturbation and the update layer by layer, eliminating the need to retain the full gradient estimator $U_\ell V_\ell^T$. Since $r_\ell \ll \min\{m_\ell, n_\ell\}$, the additional memory required for storing $U_\ell$ and $V_\ell$ is negligible. Moreover, memory costs can be further reduced using the random seed technique (Malladi et al., 2023). Instead of storing $U_\ell$ and $V_\ell$ directly, only the random seeds $s_\ell^U$ and $s_\ell^V$ used to generate them are saved. Whenever $U_\ell$ and $V_\ell$ are needed, the seeds $s_\ell^U$ and $s_\ell^V$ are used to regenerate these matrices, thereby eliminating the need for their storage. While this approach reduces memory usage, it introduces additional floating-point operations (flops) due to the regeneration process.

**stable sampling strategy.** In the main recursion (8), the variable $X^t$ is updated within the subspace spanned by $U^t$ and $V^t$ at each iteration $t$. However, if $U^t$ and $V^t$ are resampled at every iteration, the subspace will shift too frequently. This limits the algorithm's ability to adequately explore one low-rank subspace over a longer period, potentially causing abrupt changes in the model parameters $X$ at each iteration and degrading fine-tuning performance.

Additionally, ZO methods capture less information about the true gradient compared to FO algorithms, necessitating more iterations to achieve similar performance. In other words, multiple ZO steps may be required to match the progress of a single FO step. This suggests that maintaining a low-rank structure in the gradient estimator at each step is insufficient; instead, the cumulative sum of gradient estimators over several consecutive iterations must also preserve a low-rank structure.

The motivations outlined above lead us to propose a stable sampling strategy. While $U$ is sampled at every iteration $t$, we only sample $V$ every $\nu$ iterations, where $\nu > 0$ represents the chosen period duration. During the iterations $t \in \{k\nu, \dots, (k+1)\nu-1\}$ for each period $k$, the matrix $V^{(k)}$ remains fixed, thus restricting the model update to the subspace spanned by $V^{(k)}$. This leads to our proposed HZO algorithm, whose update rule for any $t \geq 0$ is defined as:

$$\left(\boldsymbol{X}^{t+1} = \boldsymbol{X}^t - \alpha \hat{\nabla} F\left(\boldsymbol{X}^t; \xi^t\right)\right), \quad \text{where} \quad \hat{\nabla} F\left(\boldsymbol{X}^t; \xi^t\right) = \text{LGE}\left(\boldsymbol{X}^t, \boldsymbol{U}^t, \boldsymbol{V}^{(k)}, \boldsymbol{r}, \epsilon, \xi^t\right). \tag{9}$$

With the stable sampling strategy, $\nabla F(X^t; \xi^t)$ consistently lies within the subspace determined by $V^{(k)}$ for any $t \in \{k\nu, \dots, (k+1)\nu - 1\}$. Therefore, the accumulation of the estimated gradients over these consecutive $\nu$ steps, which can be viewed as a more accurate approximation of the true gradient in a single FO step, has a rank that does not exceed $r$. When $\nu = 1$, the HZO update rule (9) reduces to the standard recursion (8). Detailed mathematical proof process can be found in Appendix A.1.

**Hyperparameter tuning.** The parameter $\nu$, which defines the number of steps over which $X^t$ is updated within the same subspace, is critical for performance and should be set to a moderate value. If $\nu$ is too small, frequent subspace shifts may lead to abrupt model changes, while a $\nu$ that is too large limits the algorithm to focus only on a few subspaces, potentially reducing generalization. The parameter $r$ defines the rank of the gradient estimator. Since the true gradient rank is unknown, we typically set $r$ as a small constant that is significantly less than both $m_\ell$ and $n_\ell$ to avoid additional memory overhead. In our experiments, we set $r_\ell = r$ through all layers. The typical choices for the parameters are $r = 2, 4, 8$ and $\nu = 50, 100$.

## 3.4 INTER-LAYER HYBRID OPTIMIZATION

While MeZO provides advantages in terms of memory efficiency, it experiences notable reductions in accuracy relative to complete fine-tuning using SGD or Adam. To address these limitations, we introduce a low-order hybrid optimizer designed to enhance MeZO's effectiveness while adhering to memory constraints, achieved through the fusion of a ZO optimizer and a FO optimizer. The inter-layer approach features two variations: (1) Z+F, which applies the ZO optimizer for training the shallow layers and the FO optimizer for the deep layers; (2) F+Z, which employs the FO optimizer for the shallow layers and the ZO optimizer for the deep layers.

---

**Algorithm 1** Resample-Triggered FO: HZO with FO on $t \bmod \nu = 0$

---

**Require:** Parameters $X = \{X_\ell\}_{\ell=1}^{L}$, loss $F(X; \xi)$; total steps $T$; ZO scale $\varepsilon$; ZO lr $\alpha$; FO lr $\eta$;
   stable interval $\nu$; ranks $\{r_\ell\}$; layer partition: $\mathcal{Z}$ (ZO layers), $\mathcal{O}$ (FO layers near output)
1: **for** $t = 0, \ldots, T-1$ **do**
2:     **for** $\ell \in \mathcal{Z}$ **do**
3:         Sample $U_\ell \sim \mathcal{N}(0,1)^{m_\ell \times r_\ell}$                              ▷ ZO uses fresh $U$ every step
4:         **if** $t \bmod \nu = 0$ **then**
5:             Sample $V_\ell \sim \mathcal{N}(0,1)^{n_\ell \times r_\ell}$              ▷ stable resampling of $V$ every $\nu$ steps
6:     $X \leftarrow \text{PERTURBZ}(X, +\varepsilon, \{U_\ell, V_\ell\}_{\ell \in \mathcal{Z}})$
7:     $F^+ \leftarrow F(X; \xi)$
8:     $X \leftarrow \text{PERTURBZ}(X, -2\varepsilon, \{U_\ell, V_\ell\}_{\ell \in \mathcal{Z}})$
9:     $F^- \leftarrow F(X; \xi)$
10:    $X \leftarrow \text{PERTURBZ}(X, +\varepsilon, \{U_\ell, V_\ell\}_{\ell \in \mathcal{Z}})$                    ▷ reset to base
11:    $c \leftarrow \dfrac{F^+ - F^-}{2\varepsilon}$                           ▷ two-point finite difference
12:    **for** $\ell \in \mathcal{Z}$ **do**             ▷ ZO update in the current low-rank subspace
13:       $X_\ell \leftarrow X_\ell - \alpha\, c\left(U_\ell V_\ell^\top / r_\ell\right)$
14:    **if** $t \bmod \nu = 0$ **then**   ▷ when $V$ is resampled, do one FO update on output-proximal layers
15:       $\{g_\ell\}_{\ell \in \mathcal{O}} \leftarrow \text{FO-BACKWARD}(X, \xi, \mathcal{O})$
16:       **for** $\ell \in \mathcal{O}$ **do**
17:          $X_\ell \leftarrow X_\ell - \eta\, g_\ell$           ▷ or use your optimizer: Adam/Adafactor, etc.

18: **procedure** $\text{PERTURBZ}(X, \varepsilon, \{U_\ell, V_\ell\}_{\ell \in \mathcal{Z}})$
19:     **for** $\ell \in \mathcal{Z}$ **do**
20:         $X_\ell \leftarrow X_\ell + \varepsilon U_\ell V_\ell^\top$                    ▷ perturb only ZO layers
21:     **return** $X$
22: **procedure** $\text{FO-BACKWARD}(X, \xi, \mathcal{O})$
23:     **Freeze** layers in $\mathcal{Z}$; **enable grads** for layers in $\mathcal{O}$
24:     $G \leftarrow \nabla_{\{X_\ell : \ell \in \mathcal{O}\}} F(X; \xi)$            ▷ one forward + one backward
25:     **return** $\{G_\ell\}_{\ell \in \mathcal{O}}$

---

Interlayer mixing optimization, after analyzing the peak memory consumption during the fine-tuning period, it is evident that the positioning of the FO optimization layer significantly affects overall memory usage. Compared to the F+Z solution, the Z+F solution uses the ZO optimizer for shallow layers and the FO optimizer for deep layers, which can reduce more memory consumption because they need to cache more activations. Therefore, we adopted this solution, as shown on the left side of Figure 1. In this solution, the gradient calculations of the ZO and FO optimizers are independent. For the deep layers optimized by the FO optimizer, we perform backpropagation to obtain their gradients. For the shallow layers trained using the ZO optimizer, we conduct two forward passes to approximate their gradients. Subsequently, we adjust the parameters of the shallow and deep layers independently.

**Triggering Mechanism (Resample-Triggered FO)** *The first-order (FO) update is only triggered when $t \bmod \nu = 0$, and it aligns strictly with the resampling of $V$; all other iterations perform zero-order (ZO) updates.* Specifically, within each low-rank subspace determined by the current $V$, $\nu$ consecutive ZO two-point differences and updates are performed; when $t \equiv 0 \pmod{\nu}$ and $V$ needs to be resampled, after completing the ZO updates for that step (or alternatively, first FO and then ZO), a single FO backpropagation update is performed on the layers near the output, i.e., the set $\mathcal{O}$. This strategy implements the concept of "accumulating within the subspace + fine-tuning upon switching" and is consistent with the stable sampling motivation of HZO as well as the main loop structure in Algorithm 1. It maintains stable accumulation within the subspace while quickly correcting during the subspace switch using first-order information.

### 3.5 DYNAMIC SCHEDULER DESIGN FOR EFFICIENT OVERLAP

As shown on the right side of Figure 1.We enhance memory management through the pre-allocation of a reusable memory block for transformers on the GPU, which avoids the inefficiencies associated with frequent memory allocations and deallocations amid data exchanges between the CPU and GPU. This allocated memory is flexibly repurposed for each transformer block successively, accelerating data movements and maintaining consistent GPU memory consumption, ultimately boosting overall computational performance.

Additionally, we implement the approach described by Li et al. Li et al. (2020), utilizing communication buckets to improve the efficiency of block-based transmissions. In particular, we merge parameter segments inside blocks into unified memory buckets, thereby elevating the effectiveness of communications.

To overlap the data loading and computation process, we propose a dynamic scheduler, utilizing the asynchronous execution on different CUDA streams. Specifically, our scheduler includes three CUDA streams, which are utilized to control the $i$-th transformer block's computation, the $(i+1)$-th block's uploading, and the $(i-1)$-th block's offloading concurrently. This design minimizes data transfer conflicts and maximizes GPU utilization by keeping computational and communication channels active.

A key benefit of our framework lies in the dual-forward mechanism, which effectively doubles the computation duration while leaving the communication time per block unaltered. This improvement markedly elevates the probability of achieving full overlap between communication and computation operations, particularly given that CPU-GPU communication is typically slower than GPU-based computation. Our subsequent assessments reveal that, owing to ZO's distinctive dual forward passes—which prolong computation relative to the single forward pass—communication latencies cease to represent the dominant bottleneck across the majority of cases.

## 4 EXPERIMENTS

Our experiments assess the algorithms using language models (LMs) across different sizes, encompassing RoBERTa-largeLiu et al. (2019) and extensive autoregressive LMs like OPT-13B, 30B, and 66B Zhang et al. (2022),Details can be found in the appendix A.3.1. Additionally, we examine LLaMA models Touvron et al. (2023) in multiple scales, Details can be found in the appendix A.3.3 We benchmark our HZO algorithms against MeZO Malladi et al. (2023b) and its derivatives LoHOChen et al. (2025), plus other reference methods such as zero-shot and in-context learning (ICL) techniques. Additionally, we evaluate full fine-tuning and LoRA employing the gradient-based Adam optimizer Kingma & Ba (2014), denoted as FT and FT-LoRA, respectively.To ensure equitable comparisons, we execute a comprehensive grid search over the hyperparameters specified in Malladi et al. (2023b) and adopt the superior outcomes for MeZO and its variants.

### 4.1 RESULTS ON OPT

To further assess the efficacy of HZO on extensive language models, we broaden our investigation to the OPT models Zhang et al. (2022) featuring billions of parameters (13B, 30B, and 66B).

For OPT, we conduct experiments on the following datasets: SST-2, RTE, CB De Marneffe et al. (2019), BoolQ Clark et al. (2019), WSC Levesque et al. (2012), WiC Pilehvar & Camacho-Collados (2018), MultiRC Khashabi et al. (2018), COPA Roemmele et al. (2011), ReCoRD Zhang et al. (2018), SQuAD Rajpurkar et al. (2016), and DROP Dua et al. (2019). Based on Table 2, HZO delivers strong results across multiple tasks, often approaching or surpassing full fine-tuning. On BoolQ, HZO reaches 73.04, more than six points higher than in-context learning (66.72) and above MeZO (68.02), although LoHO attains 73.68. On CoPA, HZO achieves 89.70, clearly outperforming LoHO (84.13) and full fine-tuning (78.98). For reading comprehension, HZO obtains 86.00 on SQuAD and 63.12 on MultiRC, exceeding LoHO (82.29 and 61.40, respectively); on ReCoRD, it scores 81.15—below ICL (82.29) and LoHO (81.67) but above full fine-tuning (74.28). Full fine-tuning remains stronger on several tasks, notably CB, where it reaches 84.18 while HZO attains 68.42; the gaps are small on SST-2 (91.94 for full fine-tuning and 91.76 for HZO) and RTE (70.76 for full fine-tuning and 70.21 for HZO). HZO underperforms on DROP (30.63) relative to MeZO

| Experiment | Hyperparameters | Values |
|---|---|---|
| **HZO** | Batch size | 16 |
| | Learning rate | $\{1 \times 10^{-6}, 1 \times 10^{-7}\}$ |
| | $\epsilon$ | $\{1 \times 10^{-3}, 1 \times 10^{-4}\}$ |
| | Rank (r) | $\{1, 2, 4\}$ |
| | Interval ($\nu$) | $\{50, 100\}$ |
| **MeZO** | Batch size | 16 |
| | Learning rate | $\{1 \times 10^{-6}, 1 \times 10^{-7}\}$ or $\{1 \times 10^{-6}, 5 \times 10^{-7}, 1 \times 10^{-7}\}$ for SQuAD and DROP |
| | $\epsilon$ | $1 \times 10^{-3}$ |
| **LoHO** | Batch size | 16 |
| | Learning rate | $\{1 \times 10^{-4}, 5 \times 10^{-5}\}$ or $\{1 \times 10^{-4}, 5 \times 10^{-5}, 1 \times 10^{-5}\}$ for SQuAD and DROP |
| | $\epsilon$ | $1 \times 10^{-2}$ |
| **FT** | Batch size | 8 |
| | Learning rate | $\{1 \times 10^{-5}, 5 \times 10^{-5}, 8 \times 10^{-5}\}$ |

Table 1: The hyperparameter grids used for OPT experiments

(31.13), LoHO (31.33), and full fine-tuning (31.28). Overall, HZO is a competitive alternative to full fine-tuning, with clear strengths on WSC, CoPA, and SQuAD, and competitive results on ReCoRD. HZO demonstrates accelerated convergence across different model scales, such as 13B and 66B. As

| Task | SST-2 | RTE | CB | BoolQ | WSC | WiC | MultiRC | CoPA | ReCoRD | SQuAD | DROP |
|---|---|---|---|---|---|---|---|---|---|---|---|
| Zero-shot | 58.86 | 59.45 | 46.31 | 59.15 | 38.44 | 54.83 | 47.08 | 80.03 | 81.01 | 46.23 | 14.46 |
| ICL | 87.14 | 62.18 | 57.16 | 66.72 | 39.34 | 50.39 | 52.96 | 87.10 | **82.29** | 75.94 | 29.55 |
| MeZO | 91.41 | 68.09 | 65.91 | 68.02 | 61.70 | 60.23 | 59.55 | 88.19 | 81.36 | 81.84 | 31.13 |
| LoHO | 89.71 | 68.09 | 67.85 | **73.68** | 63.43 | 60.12 | 61.40 | 84.13 | 81.67 | 82.29 | **31.33** |
| HZO | **91.76** | **70.21** | **68.42** | 73.04 | **63.81** | **63.93** | **63.12** | **89.70** | 81.15 | **86.00** | 30.63 |
| FT | 91.94 | 70.76 | 84.18 | 77.01 | 63.37 | 70.29 | 71.29 | 78.98 | 74.28 | 84.90 | 31.28 |

Table 2: Experiments on OPT-13B (with 10000 examples). ICL: in-context learning; FT: full fine-tuning with Adam. The best results are shown in bold except for FT.

shown in Figure 2, the method consistently converges faster on diverse datasets and architectures. For instance, on the WIC dataset with the OPT-66B configuration, the HZO algorithm reaches the same training loss as MeZO in only half the epochs, while also producing smoother trajectories with reduced loss oscillations.

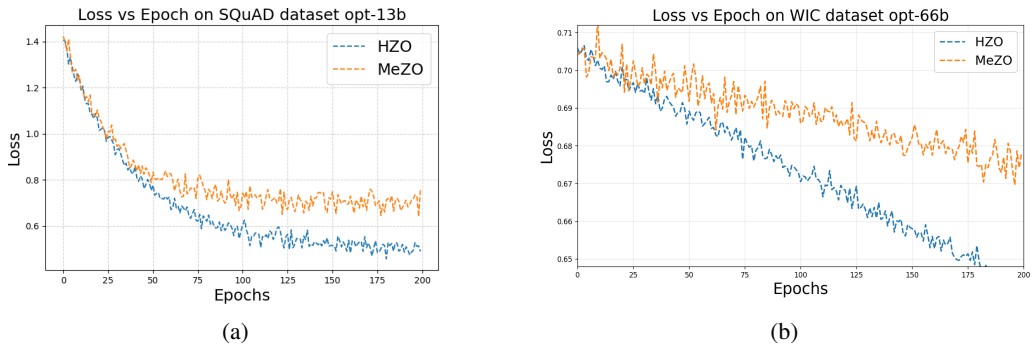

(a)

(b)

Figure 2: Convergence speed on different models and datasets

Based on Table 3, HZO shows strong performance on SuperGLUE for both OPT-30B and OPT-66B, often matching or exceeding baselines, with a few exceptions. For OPT-30B, HZO achieves 93.81 on SST-2 and 72.99 on BoolQ, outperforming in-context learning (66.24) and MeZO (68.11) on the latter. On WSC and WiC, HZO obtains 65.43 and 58.31, higher than MeZO (63.36 and 56.46) and in-context learning (56.65 and 51.24). On SQuAD, HZO reaches 85.00, above in-context learning (77.82) and zero-shot (46.70), while MeZO attains 85.97. For OPT-66B, HZO attains 92.41 on SST-2 and 74.74 on RTE, slightly above MeZO (92.14 and 71.40) and clearly above in-context learning (89.20 and 65.29). HZO also delivers 74.71 on BoolQ and 84.26 on SQuAD, exceeding MeZO (73.64 and 83.85) and in-context learning (62.61 and 81.12). The only metric where MeZO leads

| Task | SST-2 | RTE | BoolQ | WSC | WiC | SQuAD |
|---|---|---|---|---|---|---|
| 30B Zero-shot | 56.69 | 51.98 | 39.12 | 38.50 | 50.17 | 46.70 |
| 30B ICL | 82.01 | **66.82** | 66.24 | 56.65 | 51.24 | 77.82 |
| 30B MeZO | 90.67 | 64.41 | 68.11 | 63.36 | 56.46 | **85.97** |
| 30B HZO | **93.81** | 65.83 | **72.99** | **65.43** | **58.31** | 85.00 |
| 66B Zero-shot | 57.53 | 67.34 | 66.68 | 43.26 | 50.72 | 47.96 |
| 66B ICL | 89.20 | 65.29 | 62.61 | 52.90 | 55.00 | 81.12 |
| 66B MeZO | 92.14 | 71.40 | 73.64 | **64.45** | 57.67 | 83.85 |
| 66B HZO | **92.41** | **74.74** | **74.71** | 64.32 | **59.49** | **84.26** |

Table 3: Experiments on OPT-30B and OPT-66B on SuperGLUE benchmark. For each model size, the best results are shown in bold.

| | GPU Memory Usage (GB) ↓ | | | | Throughput (tokens/sec) ↑ | | | |
|---|---|---|---|---|---|---|---|---|
| Model | MeZo (FP32) | HZO(FP32) | MeZo (FP16) | HZO (FP16) | MeZo (FP32) | HZO(FP32) | MeZo (FP16) | HZO(FP16) |
| OPT-1.3B | 8.69 | 4.98(x0.57) | 5.68(x0.65) | **3.64(x0.42)** | 1998 | 1955(x0.97) | **6629(x3.32)** | 6448(x3.23) |
| OPT-2.7B | 14.17 | 5.79(x0.41) | 8.84(x0.62) | **4.04(x0.29)** | 1104 | 1086(x0.98) | **4229(x3.83)** | 4220(x3.82) |
| OPT-6.7B | 32.16 | 8.22(x0.26) | 16.20(x0.50) | **4.88(x0.15)** | 492 | 485(x0.98) | **2349(x4.77)** | 2270(x4.61) |
| OPT-13B | 57.38 | 10.48(x0.18) | 28.99(x0.51) | **6.04(x0.11)** | 266 | 259(x0.97) | **1326(x4.98)** | 1251(x4.70) |
| OPT-30B | 143.45 | 15.61(x0.11) | 62.40(x0.43) | **8.65(x0.06)** | 136 | 122(x0.90) | **641(x4.71)** | 514(x3.78) |
| OPT-66B | 290.75 | 21.77(x0.07) | 126.33(x0.43) | **11.79(x0.04)** | 46 | 40(x0.87) | **325(x7.07)** | 273(x5.93) |
| OPT-175B | 623.49 | 33.22(x0.05) | 269.27(x0.43) | **17.62(x0.03)** | 17 | 14(x0.82) | **56(x3.29)** | 37(x2.18) |

Table 4: Key findings from HZO performance across diverse model setups and in both FP32 and FP16 precision modes. Bold numbers indicate the lowest memory usage and highest throughput for each model. Our HZO achieves a throughput close to MeZO while maintaining minimal memory consumption.

is WSC (64.45 compared with HZO 64.32). Overall, HZO provides robust gains over in-context learning and zero-shot, and often matches or slightly outperforms MeZO across large models.

Based on Table 4, HZO consistently reduces GPU memory usage across OPT models. For OPT-6.7B, it uses 8.22 GB in FP32 compared with MeZO's 32.16 GB (reduction $\approx 74\%$) and 4.88 GB in FP16 compared with 16.20 GB (reduction $\approx 70\%$). These reductions enable larger models: in FP16, HZO requires 11.79 GB for OPT-66B and 17.62 GB for OPT-175B, whereas MeZO needs 126.33 GB and 269.27 GB. Throughput remains broadly comparable to MeZO: on OPT-2.7B in FP32, HZO achieves 1086 tokens/s compared with 1104; in FP16 it is typically within about 3–20% of MeZO (for OPT-66B, 273 tokens/s compared with 325). Overall, FP16 memory savings range from roughly 36–54% on smaller models to about 70–95% from 6.7B upward, while maintaining throughput close to MeZO.

## 4.2 ABLATION STUDY

Details of the ablation experiment can be found in Appendix A.3.4. We have also proven the convergence, with details in Appendix A.2.

## 5 CONCLUSION

In summary, this work introduces a low-rank zeroth-order gradient estimation method that effectively reduces memory consumption while preserving competitive fine-tuning performance. By combining hierarchical block-wise optimization with a hybrid ZO–FO strategy and proposing an offloading scheme between GPU and CPU memory, the method enables large-scale models such as OPT-175B to be fine-tuned on limited hardware resources. The proposed approach not only mitigates the slow convergence issue inherent in conventional ZO methods but also achieves convergence behavior comparable to first-order algorithms, thereby providing a practical and efficient solution for parameter-efficient fine-tuning of large language models under memory-constrained environments.

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

# A APPENDIX

You may include other additional sections here.

## A.1 HZO IS A ZERO-ORDER SUBSPACE OPTIMIZATION METHOD

Let $F(X; \xi)$ be the stochastic objective, with $X \in \mathbb{R}^{m \times n}$ and randomness $\xi$. Fix a rank parameter $r \in \mathbb{N}$ with $r \leq \min\{m, n\}$, a perturbation radius $\varepsilon > 0$, and, within epoch $k$ of length $\nu$, a direction

matrix $V^{(k)} \in \mathbb{R}^{n \times r}$ that is held fixed. For any $X \in \mathbb{R}^{m \times n}$, $U \in \mathbb{R}^{m \times r}$, $V \in \mathbb{R}^{n \times r}$, define the low-rank zeroth-order gradient estimator

$$\mathrm{LGE}(X, U, V, r, \varepsilon, \xi) := \frac{F(X + \varepsilon\, UV^\top; \xi) - F(X - \varepsilon\, UV^\top; \xi)}{2\varepsilon} \cdot \frac{UV^\top}{r}. \tag{10}$$

**HZO update.** Within epoch $k$, for each inner step $t \in \{k\nu, \ldots, (k+1)\nu - 1\}$, HZO samples $U_t \in \mathbb{R}^{m \times r}$, keeps $V^{(k)}$ fixed, and performs

$$\mathrm{X}_{t+1} = X_t - \alpha\, \mathrm{LGE}(X_t,\, U_t,\, V^{(k)},\, r,\, \varepsilon,\, \xi_t) = X_t - \alpha \cdot \frac{F(X_t + \varepsilon\, U_t (V^{(k)})^\top; \xi_t) - F(X_t - \varepsilon\, U_t (V^{(k)})^\top; \xi_t)}{2\varepsilon} \cdot \frac{U_t (V^{(k)})^\top}{r}. \tag{11}$$

**Zeroth-order subspace method.** In the same epoch, consider the subspace iterate anchored at $\widetilde{X}^{(k)} \in \mathbb{R}^{m \times n}$ and parameterized by $B \in \mathbb{R}^{m \times r}$. Starting from $B^{(k,0)} = \mathbf{0}$, perform $\nu$ inner steps

$$B^{(k,s+1)} = B^{(k,s)} - \gamma\, \widehat{\nabla}_B F(\widetilde{X}^{(k)} + B^{(k,s)} (V^{(k)})^\top; \xi^{(k,s)}), \quad s = 0, 1, \ldots, \nu - 1, \tag{12}$$

where each inner step draws an independent $U^{(k,s)} \in \mathbb{R}^{m \times r}$ and uses the symmetric-difference zeroth-order estimator (applied to the $B$ variable)

$$\widehat{\nabla}_B F(Z; \xi) = \frac{F(Z + \varepsilon\, U^{(k,s)} (V^{(k)})^\top; \xi) - F(Z - \varepsilon\, U^{(k,s)} (V^{(k)})^\top; \xi)}{2\varepsilon}\, U^{(k,s)}. \tag{13}$$

**Lifting to the ambient space.** Define, for each inner step,

$$Y^{(k,s)} := \widetilde{X}^{(k)} + B^{(k,s)} (V^{(k)})^\top \in \mathbb{R}^{m \times n}.$$

Then

$$
\begin{aligned}
Y^{(k,s+1)} - Y^{(k,s)} &= (B^{(k,s+1)} - B^{(k,s)})(V^{(k)})^\top \\
&= -\gamma \frac{F(\widetilde{X}^{(k)} + (B^{(k,s)} + \varepsilon U^{(k,s)})(V^{(k)})^\top; \xi^{(k,s)}) - F(\widetilde{X}^{(k)} + (B^{(k,s)} - \varepsilon U^{(k,s)})(V^{(k)})^\top; \xi^{(k,s)})}{2\varepsilon}\, U^{(k,s)}(V^{(k)})^\top \\
&= -\gamma \frac{F(Y^{(k,s)} + \varepsilon U^{(k,s)}(V^{(k)})^\top; \xi^{(k,s)}) - F(Y^{(k,s)} - \varepsilon U^{(k,s)}(V^{(k)})^\top; \xi^{(k,s)})}{2\varepsilon}\, U^{(k,s)}(V^{(k)})^\top.
\end{aligned}
\tag{14}
$$

**Step-size coupling and identification.** Choose $\gamma = \alpha/r$. Substituting yields

$$
\begin{aligned}
Y^{(k,s+1)} &= Y^{(k,s)} - \alpha \cdot \frac{F(Y^{(k,s)} + \varepsilon U^{(k,s)}(V^{(k)})^\top; \xi^{(k,s)}) - F(Y^{(k,s)} - \varepsilon U^{(k,s)}(V^{(k)})^\top; \xi^{(k,s)})}{2\varepsilon} \cdot \frac{U^{(k,s)}(V^{(k)})^\top}{r} \\
&= Y^{(k,s)} - \alpha\, \mathrm{LGE}(Y^{(k,s)}, U^{(k,s)}, V^{(k)}, r, \varepsilon, \xi^{(k,s)}),
\end{aligned}
\tag{15}
$$

which is exactly the HZO inner-step update executed at the state $Y^{(k,s)}$ with the same fixed $V^{(k)}$ and the same sampled $U^{(k,s)}$.

**Trajectory equivalence.** Assume a shared initialization $X_0 = \widetilde{X}^{(0)}$ and, at the start of epoch $k$, $X_{k\nu} = \widetilde{X}^{(k)}$. Because the one-step updates coincide under the same randomness within the epoch, induction over $s = 0, 1, \ldots, \nu$ gives $Y^{(k,s)} = X_{k\nu+s}$. In particular,

$$\widetilde{X}^{(k+1)} = \widetilde{X}^{(k)} + B^{(k,\nu)} (V^{(k)})^\top = Y^{(k,\nu)} = X_{(k+1)\nu}. \tag{16}$$

**Conclusion.** Holding $V^{(k)}$ fixed within each epoch and coupling step sizes by $\gamma = \alpha/r$, HZO is trajectory-wise equivalent to performing $\nu$ steps of standard zeroth-order SGD on the low-rank subspace spanned by $V^{(k)}$, followed by writing the increment back to the full space.

## A.2 CONVERGENT PROOF

[Convergence of HZO] Let $f(X) = \mathbb{E}_\xi[F(X; \xi)]$ with $X = \{X_\ell\}_{\ell=1}^{L}$ and assume: (i) for every $\xi$, $F(\cdot; \xi)$ is differentiable and has $L$-Lipschitz gradient; (ii) $\mathbb{E}[\nabla F(X; \xi)] = \nabla f(X)$ and $\mathbb{E}\|\nabla F(X; \xi) - \nabla f(X)\|^2 \leq \sigma^2$; (iii) at each period the column matrix $V = \{V_\ell\}$ satisfies $V_\ell^\top V_\ell = n_\ell I$ and $\mathbb{E}[V_\ell V_\ell^\top] = r_\ell I$. Consider HZO with rank $r = \{r_\ell\}$, stable resampling interval

$\nu$, stepsize $\alpha$, and smoothing radius $\varepsilon$. Then, for $T = K\nu$ iterations and a suitable $\alpha \leq \frac{1}{144\, Lmn\nu}$ (with $m = \sum_\ell m_\ell$, $n = \sum_\ell n_\ell$), the iterates $\{X_t\}$ satisfy

$$\frac{1}{K} \sum_{k=0}^{K-1} \mathbb{E}\big\|\nabla f(X_{k\nu})\big\|^2 \;\leq\; \frac{8\Delta_0}{T\alpha} \;+\; \frac{56\, L\, m\, n^2\, \sigma^2}{r}\, \alpha \;+\; \frac{16\, L\, m\, n\, r\, \varepsilon^2}{\nu\, \alpha},$$

where $\Delta_0 = f(X_0) - f$. With the choices

$$\varepsilon = \sqrt{\frac{\Delta_0\, \nu}{16\, T\, L\, m\, n\, r}}, \qquad \alpha = \Big(144\, L\, m\, n\, \nu \;+\; \sqrt{\frac{56\, T\, L\, m\, n^2\, \sigma^2}{9\, \Delta_0\, r}}\Big)^{-1},$$

it follows that

$$\frac{1}{K} \sum_{k=0}^{K-1} \mathbb{E}\big\|\nabla f(X_{k\nu})\big\|^2 \;\leq\; 16\sqrt{\frac{7\, \Delta_0\, L\, m\, n^2\, \sigma^2}{r\, T}} \;+\; \frac{2592\, \Delta_0\, L\, m\, n\, \nu}{T}.$$

Equivalently, aggregating across layers with $d = \sum_\ell m_\ell n_\ell$ and $\tilde{d} = \sum_\ell m_\ell n_\ell^2/r_\ell$ gives the big-O rate

$$\frac{1}{K} \sum_{k=0}^{K-1} \mathbb{E}\big\|\nabla f(X_{k\nu})\big\|^2 = \mathcal{O}\left(\sqrt{\frac{\Delta_0\, L\, \tilde{d}\, \sigma^2}{T}} + \frac{\Delta_0\, L\, d\, \nu}{T}\right).$$

**Step 1 (Low-rank ZO estimator and unbiasedness).** For layer $\ell$, sample $U_\ell \in \mathbb{R}^{m_\ell \times r_\ell}$ and $V_\ell \in \mathbb{R}^{n_\ell \times r_\ell}$, and define the matrix-wise finite-difference estimator

$$\widehat{\nabla}_{X_\ell} F(X;\xi) \;:=\; \frac{F(\{X_j + \varepsilon U_j V_j^\top\}_{j=1}^L; \xi) - F(\{X_j - \varepsilon U_j V_j^\top\}_{j=1}^L; \xi)}{2\varepsilon} \cdot \frac{U_\ell V_\ell^\top}{r_\ell}.$$

As $\varepsilon \to 0$, the estimator is asymptotically unbiased in expectation over the Gaussian directions, i.e. $\lim_{\varepsilon \to 0} \mathbb{E}[\widehat{\nabla}_{X_\ell} F(X;\xi)] = \nabla_{X_\ell} F(X)$ for every $\ell$. Summing over $\ell$ gives an unbiased matrix-wise gradient in the limit.

**Step 2 (Equivalence to ZO subspace optimization).** Fix $V$, and write the subproblem $g_{X,V}(B) = \mathbb{E}_\xi[F(X + BV^\top; \xi)]$ in the "low-rank variable" $B = \{B_\ell\}$ (with $B_\ell \in \mathbb{R}^{m_\ell \times r_\ell}$). Running $\nu$ steps of standard ZO-SGD on $g_{X,V}$ with stepsize $\gamma$ and then updating $X \leftarrow X + B_\nu V^\top$ is equivalent to HZO provided $\gamma = \alpha/r$; more precisely, if both methods start from the same initialization, then $X_{k\nu} = \widetilde{X}^{(k)}$ for all periods $k$, where $\widetilde{X}^{(k)}$ denotes the outer iterate of the subspace method. Hence it suffices to prove descent on $g_{X,V}$ over a period and translate it back to $f$.

**Step 3 (Second moment bound and Gaussian smoothing).** Let $G_{X,V}(B;\xi) = F(X + BV^\top; \xi)$ and denote the ZO estimator for $\nabla_B G$ by

$$\widehat{\nabla} G_{X,V}(B;\xi) \;:=\; \frac{G_{X,V}(B + \varepsilon U; \xi) - G_{X,V}(B - \varepsilon U; \xi)}{2\varepsilon}\, U,$$

with $U$ having i.i.d. standard normal entries and overall rank parameter $mr = \sum_\ell m_\ell r_\ell$. Under the smoothness assumption, its second moment admits

$$\mathbb{E}\big\|\widehat{\nabla} G_{X,V}(B;\xi)\big\|_F^2 \;\leq\; 6\, m\, r\, \big\|\nabla G_{X,V}(B;\xi)\big\|_F^2 \;+\; 64\, \widetilde{L}^{\,2}\, m^3 r^3\, \varepsilon^2,$$

for an appropriate smoothness constant $\widetilde{L}$ induced by the reparameterization through $V$. Introduce the Gaussian-smoothed subproblem $g_\varepsilon(B) = \mathbb{E}_U[g_{X,V}(B + \varepsilon U)]$. Then $\mathbb{E}_{U,\xi}[\widehat{\nabla} G_{X,V}(B;\xi)] = \nabla g_\varepsilon(B)$, $g_\varepsilon$ is $\widetilde{L}$-smooth, and the smoothing biases satisfy $\big|g_\varepsilon(B) - g_{X,V}(B)\big| \leq \frac{1}{2}\widetilde{L}\, mr\, \varepsilon^2$ and $\|\nabla g_\varepsilon(B) - \nabla g_{X,V}(B)\|_F^2 \leq \widetilde{L}^{\,2} mr\varepsilon^2$.

**Step 4 (One-period expected descent on the subproblem).** Run the $\nu$ inner ZO-SGD steps on $g_{X,V}$ with stepsize $\alpha/r$. By $\widetilde{L}$-smoothness of $g_\varepsilon$, the unbiasedness property in Step 3, and the second-moment bound, summing the standard descent inequality over a period and choosing $\alpha$ small enough yields

$$\mathbb{E}\big[g_{X,V}(B_\nu) - g_{X,V}(B_0)\big] \;\leq\; -\Big(\tfrac{\nu\alpha}{4r} - c_1\, \nu\, \alpha^2\Big) \mathbb{E}\|\nabla g_{X,V}(B_0)\|_F^2 \;+\; c_2\, \nu\, \alpha^2\, \sigma^2 \;+\; c_3\, m\, r\, \varepsilon^2,$$

for absolute constants $c_1, c_2, c_3$ that depend on $L, \widetilde{L}, m$ but not on $T$. When $\alpha$ satisfies the stated small-stepsize condition, the bracketed coefficient is positive and we obtain a net expected decrease dominated by $\|\nabla g_{X,V}(B_0)\|_F^2$, up to variance and smoothing terms.

**Step 5 (Back to the full problem and telescoping).** Because $g_{X_k,V^{(k)}}(B_\nu) = f(X_{(k+1)\nu})$ and $g_{X_k,V^{(k)}}(B_0) = f(X_{k\nu})$, the inequality in Step 4 becomes a per-period reduction bound for $f$. Taking expectation over the fresh $V^{(k)}$ and using $\mathbb{E}[V^{(k)}(V^{(k)})^\top] = I$ gives $\mathbb{E}\|\nabla g_{X_k,V^{(k)}}(B_0)\|_F^2 = \mathbb{E}\|\nabla f(X_{k\nu})\|_F^2$. Summing the per-period inequality over $k = 0, \ldots, K-1$ (so $T = K\nu$) telescopes the function values and yields

$$\frac{1}{K} \sum_{k=0}^{K-1} \mathbb{E}\big\|\nabla f(X_{k\nu})\big\|^2 \ \leq \ \frac{8\Delta_0}{T\alpha} \ + \ \frac{56\, L\, m\, n^2\, \sigma^2}{r}\, \alpha \ + \ \frac{16\, L\, m\, n\, r\, \varepsilon^2}{\nu\, \alpha},$$

under $\alpha \leq \frac{1}{144\, Lmn\nu}$. Optimizing the bound over $(\alpha, \varepsilon)$ by balancing the variance and smoothing terms gives the stated parameter choices and the rate

$$\frac{1}{K} \sum_{k=0}^{K-1} \mathbb{E}\big\|\nabla f(X_{k\nu})\big\|^2 \ \leq \ 16 \sqrt{\frac{7\, \Delta_0\, L\, m\, n^2\, \sigma^2}{r\, T}} \ + \ \frac{2592\, \Delta_0\, L\, m\, n\, \nu}{T}.$$

Finally, summing the per-layer bounds and recalling $d = \sum_\ell m_\ell n_\ell$ and $\tilde{d} = \sum_\ell m_\ell n_\ell^2 / r_\ell$ yields the big-O statement in the theorem.

### A.3 MORE EXPERIMENTS

#### A.3.1 RESULTS ON ROBERTA-LARGE

For RoBERTa-large, we evaluate the performance on six NLP tasks: SST-2 Socher et al. (2013), SST-5 Socher et al. (2013), SNLI Bowman et al. (2015), MNLI Williams et al. (2018), RTE Dagan et al. (2005); Bar-Haim et al. (2006); Giampiccolo et al. (2007); Bentivogli et al. (2009), and TREC Voorhees & Tice (2000). We adopt two settings: $k = 16$ and $k = 512$, which require 16 and 512 examples per class, respectively, during both the training and validation stages.

| Experiment | Hyperparameters | Values |
|:---:|:---:|:---:|
| **HZO** | Batch size | 64 |
| | Learning rate (k=16) | $1 \times 10^{-6}$ |
| | Learning rate (k=512) | $2 \times 10^{-7}$ |
| | Rank (r) | $\{4, 8\}$ |
| | Interval ($\nu$) | $\{50, 100\}$ |
| | $\epsilon$ | $1 \times 10^{-3}$ |
| | Weight Decay | 0 |
| **MeZO** | Batch size | 64 |
| | Learning rate | $\{1 \times 10^{-7}, 1 \times 10^{-6}, 1 \times 10^{-5}\}$ |
| | $\epsilon$ | $1 \times 10^{-3}$ |
| | Weight Decay | 0 |
| **LoHO** | Batch size | 64 |
| | Learning rate | $\{1 \times 10^{-5}, 5 \times 10^{-5}, 1 \times 10^{-4}\}$ |
| | $\epsilon$ | $1 \times 10^{-3}$ |
| | Weight Decay | 0.1 |
| **FT** | Batch size | 8 |
| | Learning rate | $\{1 \times 10^{-5}, 3 \times 10^{-5}, 5 \times 10^{-5}\}$ |
| | Weight Decay | 0 |
| **FT-LoRA** | Batch size | 8 |
| | Learning rate | $\{1 \times 10^{-4}, 3 \times 10^{-4}, 5 \times 10^{-4}\}$ |
| | Rank (r), $\alpha$ | (8, 16) |

Table 5: The hyperparameter grids used for RoBERTa-large experiments

| Task | SST-2 | SST-5 | SNLI | MNLI | RTE | TREC |
|------|-------|-------|------|------|-----|------|
| Type | sentiment | sentiment | natural language inference | natural language inference | topic | |
| Zero-shot | 79.06 | 35.39 | 50.37 | 48.63 | 51.41 | 32.16 |
| **Gradient-free methods: k = 16** | | | | | | |
| MeZO | 86.54 (7.97) | 41.93 (1.94) | 68.22 (4.50) | 56.81 (3.21) | 58.48 (9.65) | 61.80 (10.35) |
| LoHO | 88.54 (1.73) | 38.89 (2.00) | 67.09 (3.26) | 60.17 (6.67) | **59.81** (4.31) | 57.61 (5.23) |
| HZO | **89.06** (2.16) | **42.18** (1.66) | **75.18** (4.90) | **64.89** (4.66) | 58.25 (9.16) | **82.29** (5.73) |
| **Gradient-based methods: k = 16** | | | | | | |
| FT | 89.36 (5.58) | 44.25 (1.72) | 72.82 (5.55) | 66.44 (4.42) | 59.54 (4.79) | 84.59 (4.65) |
| FT-LoRA | 92.87 (0.81) | 45.47 (1.34) | 71.38 (4.55) | 61.57 (6.82) | 61.79 (4.63) | 75.97 (7.42) |
| **Gradient-free methods: k = 512** | | | | | | |
| MeZO | 92.69 (0.36) | **53.80** (2.05) | 84.25 (1.71) | 84.75 (1.27) | 71.98 (2.93) | 95.06 (0.44) |
| LoHO | 91.73 (0.33) | 45.29 (1.36) | 73.01 (0.92) | 73.23 (1.26) | 72.71 (0.68) | 94.86 (1.86) |
| HZO | **94.65** (0.22) | 53.02 (0.15) | **86.01** (1.01) | **85.25** (1.03) | **79.04** (1.62) | **95.44** (0.44) |
| **Gradient-based methods: k = 512** | | | | | | |
| FT | 96.36 (0.65) | 55.82 (1.70) | 88.48 (0.78) | 84.72 (0.65) | 82.03 (1.59) | 97.23 (0.44) |
| FT-LoRA | 91.83 (2.04) | 52.25 (1.04) | 84.41 (1.26) | 74.63 (3.38) | 81.30 (1.73) | 96.29 (0.65) |

Table 6: Experimental results on RoBERTa-large (350M). All reported numbers are averaged accuracy (standard deviation).

Based on the results in Table 6, the HZO method performs stably across tasks. Under the Zero-shot setting, HZO achieves 89.06 (SST-2), 42.18 (SST-5), and 75.18 (SNLI), outperforming other gradient-free methods. For Gradient-free methods: k = 512, HZO obtains 94.65 (SST-2), 85.25 (MNLI), 79.10 (RTE), and 95.44 (TREC), which are close to the results of full fine-tuning (FT) and FT-LoRA. While FT achieves slightly higher scores on sentiment and inference tasks, HZO remains competitive without requiring full gradient updates, and even surpasses FT and FT-LoRA on TREC. Overall, HZO shows strong and stable performance, particularly outstanding in RTE and TREC tasks, with efficiency advantages over gradient-based methods.

### A.3.2 RESULTS ON OPT

Compared to full fine-tuning, we can complete fine-tuning with MeZO using much less GPU memory, requiring only 7 GB memory and a single GPU on RTE and 6.9 GB on MultiRC.7

| Task | RTE | | MultiRC | |
|------|-----|--|---------|--|
| | Memory | Consumed GPUs | Memory | Consumed GPUs |
| HZO | 7.0 GB | $1 \times$ A800 | 6.9 GB | $1 \times$ A800 |
| MeZo | 29.0 GB | $1 \times$ A800 | 28.3 GB | $1 \times$ A800 |
| LoHO | 49.7 GB | $1 \times$ A800 | 50.1 GB | $1 \times$ A800 |
| FT-LoRA | 79.0 GB | $1 \times$ A800 | 102.4 GB | $2 \times$ A800 |
| FT | 250.0 GB | $4 \times$ A800 | 315.2 GB | $4 \times$ A800 |

Table 7: HZO achieves competitive performance with drastically lower resource consumption on OPT-13B.

### A.3.3 RESULTS ON LLAMA

As shown in the table 9, HZO demonstrates strong performance across multiple tasks and models. It excels in **SST-2** with a score of 95.8, matching the best results, and performs competitively in **COPA** (85.3) and **SQuAD** (90.3). HZO also outperforms other methods in **WG** on LLama-7B, with a score of 71.0. While its performance in **WiC** (57.9) lags behind other methods, it remains competitive in general, especially in **COPA** and **SQuAD**. Overall, HZO shows balanced and robust results, particularly in sentiment analysis and world knowledge tasks, with minor room for improvement in tasks requiring fine-grained contextual understanding.

### A.3.4 ABLATION STUDY

In this section, we investigate the influence of the rank parameter $r$ and the stable update interval $\nu$ on the performance of the proposed algorithm. Specifically, we evaluate their effects on the OPT-1.3B model using the SST-2, COPA, and RTE datasets. To demonstrate the convergence behavior under

| Experiment | Hyperparameters | Values |
|---|---|---|
| **HZO** | Batch size | 16 |
| | Learning rate (k=16) | {1e-7} |
| | Rank (r) | {2, 4} |
| | Interval ($\nu$) | {50, 100} |
| | $\epsilon$ | 1e-3 |
| | Weight Decay | 0 |
| **MeZO** | Batch size | 16 |
| | Learning rate | {1e-7, 1e-6} |
| | $\epsilon$ | 1e-3 |
| | Weight Decay | 0 |
| **FT** | Batch size | {8} |
| | Learning rate | {1e-6, 1e-7} |
| **FT-LoRA** | Batch size | {8} |
| | Learning rate | {1e-4, 1e-5} |
| | (r, $\alpha$) | (8, 16) |

Table 8: The hyperparameter grids used for LLaMA experiments. The learning rate of the HZO algorithm refers to $\alpha/r$.

| Model | LLama-7B | | | | | LLama-13B | | LLama-70B |
|---|---|---|---|---|---|---|---|---|
| | SST-2 | WiC | COPA | SQuAD | WG | SST-2 | WG | WG |
| HZO | **95.8** | 57.9 | 85.3 | 90.3 | **71.0** | 93.6 | 68.6 | 72.1 |
| MeZO | 91.8 | 56.3 | **86.0** | 90.0 | 64.3 | 92.1 | 67.2 | 72.1 |
| FT-LoRA | 95.8 | 69.4 | 84.0 | 91.2 | 70.9 | 95.5 | **76.6** | 50.4 |
| FT | 94.1 | **72.8** | 83.0 | **90.6** | 64.4 | **96.4** | 73.3 | **75** |

Table 9: Performance comparison of different models on various tasks

varying configurations, we present the loss curves across training epochs in Figure 3. Furthermore, the corresponding accuracy and training loss obtained with different combinations of $r$ and $\nu$ are summarized in Table 10.

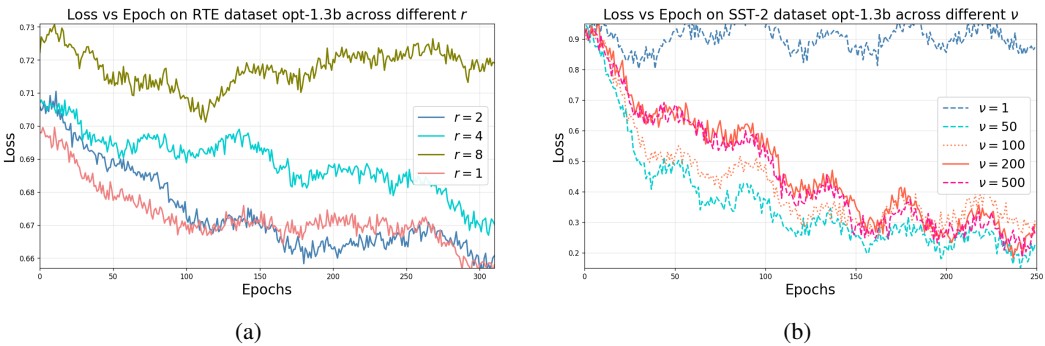

(a)                                                      (b)

Figure 3: Left: Loss curves of OPT-1.3B on RTE dataset across different rank $r$.Right: Loss curves of OPT-1.3B on SST-2 dataset across different value $\nu$.

A small rank $r$ is consistently associated with degraded model performance (Table 10, Fig. 3a). When the rank is fixed at $r = 1$, the predictive accuracy drops substantially across all datasets, suggesting a severe limitation in representational capacity. For instance, SST-2 accuracy decreases to 55.0% with a corresponding loss of 0.79, while RTE accuracy falls to 50.9% (Table 10). These results indicate that an insufficiently expressive subspace impairs the model's ability to generalize effectively.

An increased value of $\nu$ enhances convergence stability and test performance (Table 10, Fig. 3b). For SST-2, larger $\nu$ values such as $\nu = 50, 100, 200$ consistently yield accuracies above 91% with

| | | SST-2 | | COPA | | RTE | |
|---|---|---|---|---|---|---|---|
| $r$ | $\nu$ | Accuracy | loss | Accuracy | loss | Accuracy | loss |
| 1 | 50 | 88.1 | 0.45 | 72.8 | 1.93 | 56.7 | 0.68 |
| | 100 | 89.1 | 0.46 | 74.0 | 2.18 | 56.8 | 0.68 |
| 2 | 1 | 55.0 | 0.79 | 74.1 | 2.58 | 50.9 | 0.70 |
| | 50 | 93.0 | 0.37 | 74.0 | 2.04 | 61.0 | 0.69 |
| | 100 | 92.0 | 0.37 | 70.8 | 2.05 | 57.9 | 0.68 |
| | 200 | 92.9 | 0.37 | 76.9 | 2.05 | 62.1 | 0.67 |
| | 500 | 91.9 | 0.37 | 74.9 | 2.05 | 62.8 | 0.67 |
| 4 | 50 | 91.3 | 0.35 | 76.0 | 1.99 | 57.4 | 0.69 |
| | 100 | 91.9 | 0.35 | 74.8 | 1.97 | 57.7 | 0.69 |
| 8 | 50 | 88.7 | 0.48 | 71.2 | 2.03 | 55.0 | 0.71 |
| | 100 | 89.0 | 0.45 | 73.0 | 2.03 | 56.4 | 0.71 |

Table 10: Results on SST-2, COPA and RTE across different $r$ and $\nu$.

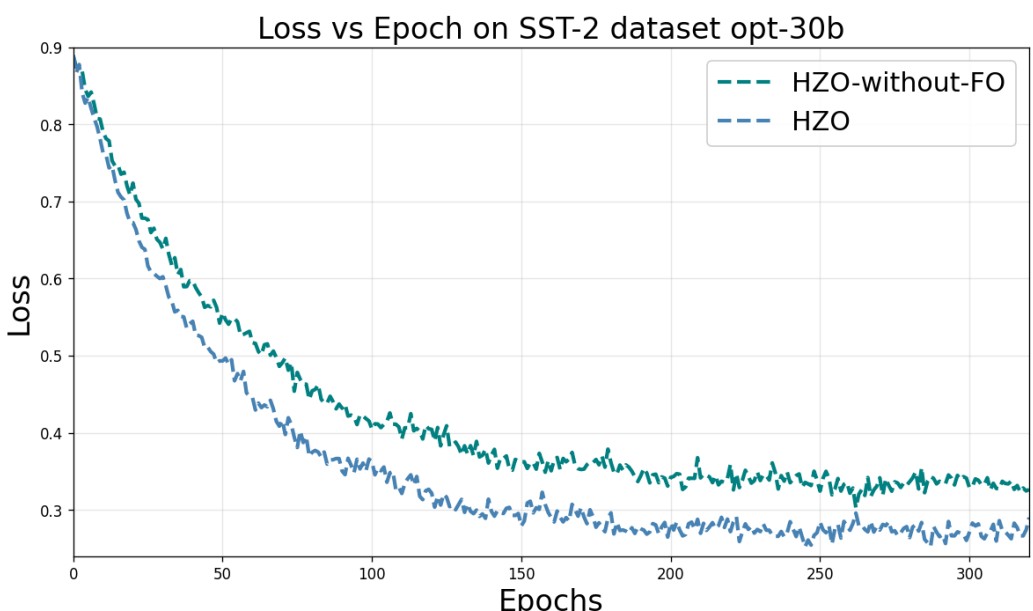

Figure 4: We tested the convergence speed of using only ZO optimization without mixed optimization, proving that mixed optimization has a significant promotion effect on convergence speed

relatively low losses around 0.37, suggesting improved optimization dynamics and stronger generalization. However, excessively large values such as $\nu = 500$ do not produce further gains, reflecting diminishing returns beyond a moderate threshold.

Moderate ranks ($r = 2$ or $r = 4$) provide the most favorable trade-off (Table 10). Compared to $r = 1$, these configurations deliver robust and balanced improvements across all benchmarks. For example, at $r = 4, \nu = 50$, SST-2 achieves 91.3% accuracy with the lowest observed loss of 0.35, while COPA and RTE maintain competitive performance. This suggests that a moderate rank effectively balances representational richness with training efficiency.

Excessively large ranks such as $r = 8$ do not guarantee additional performance benefits (Table 10). Although higher ranks may theoretically increase modeling capacity, empirical results indicate no consistent advantage. For instance, at $r = 8, \nu = 100$, COPA accuracy of 73.0% is lower than that observed at $r = 4$, implying that overly complex subspaces may introduce redundancy or hinder generalization without improving accuracy.

