# OpenReview forum: "Boosting Language Model Fine-Tuning via Zeroth-Order Hybrid Methods with Additional  Memory Aid"
_ICLR.cc/2026/Conference — Submitted to ICLR 2026_

### Official Review · Reviewer_Bu8y · 2025-10-18

**Soundness:** 2
**Presentation:** 1
**Contribution:** 1
**Rating:** 2
**Confidence:** 5

**Summary:**

This paper proposes HZO, a hybrid zeroth-order optimization framework for large language model fine-tuning. The method combines a low-rank zeroth-order gradient estimator with a hybrid update strategy, where shallow layers are updated via zeroth-order approximations while deeper layers receive occasional first-order updates. Additionally, the authors design a memory offloading and scheduling mechanism to enable fine-tuning of very large models on limited GPU memory.

**Strengths:**

- Addresses the practical and relevant challenge of fine-tuning large-scale LLMs under memory constraints.
- Experimental evaluation spans multiple datasets and large LLM(66B) and demonstrates improvements over MeZO and other zeroth-order baselines.

**Weaknesses:**

- **Limited novelty**: The low-rank idea has already been explored in HiZOO-L[1], LOZO[2], TeZO[3] which also exploit low-rank properties in zeroth-order optimization. These related methods should be explicitly cited and compared in experiments to clarify the unique contribution of this paper.

- **Model choice**: Experiments are mainly conducted on relatively older OPT models. It would strengthen the work to include evaluations on more recent LLMs to better demonstrate generality.

- **Presentation issues**: The tables are not well-organized. For example, Tables 2, 3, and 4 are messy. Table 1, which is mostly implementation detail, might be better placed in the appendix. Overall presentation should be cleaned up for readability.

- **Ablation limitations**: The first-order updates are restricted to only the final layer. Why is this the case? Would updating multiple layers yield better results? An ablation study on different layer selections is necessary.

- **Loss curves**: More training loss curves across different tasks and models should be included to demonstrate robustness and stability of the method.

[1]. Zhao, Y., Dang, S., Ye, H., Dai, G., Qian, Y., & Tsang, I. W. (2024). Second-order fine-tuning without pain for llms: A hessian informed zeroth-order optimizer. arXiv preprint arXiv:2402.15173

[2] Chen, Y., Zhang, Y., Cao, L., Yuan, K., & Wen, Z. (2024). Enhancing zeroth-order fine-tuning for language models with low-rank structures. arXiv preprint arXiv:2410.07698.

[3]Sun, Y., Huang, T., Ding, L., Shen, L., & Tao, D. (2025). TeZO: Empowering the Low-Rankness on the Temporal Dimension in the Zeroth-Order Optimization for Fine-tuning LLMs. arXiv preprint arXiv:2501.19057.

**Questions:**

Seen above.

---

### Official Review · Reviewer_Pbzi · 2025-10-20

**Soundness:** 3
**Presentation:** 2
**Contribution:** 2
**Rating:** 4
**Confidence:** 2

**Summary:**

This paper proposes Hybrid Zeroth-Order Optimization (HZO), an inter-layer hybrid optimization strategy combined with a dynamic scheduler, aiming to address the challenge of fine-tuning large language models (LLMs) under limited GPU memory. The authors claim that HZO enables fine-tuning of models as large as OPT-175B on a single 24 GB GPU. To achieve this, the paper introduces: 1. A zeroth-order gradient estimator that preserves low-rank structure to reduce computational overhead, combined with a "stable sampling strategy" to improve convergence stability. 2. An inter-layer hybrid optimization strategy, where shallow layers are optimized via zeroth-order (ZO) methods and deeper layers via first-order (FO) methods to accelerate convergence. 3. A memory scheduling and block offloading mechanism that transfers data between CPU and GPU to reduce memory footprint. Experimental comparisons are conducted between HZO, MeZO, LoHO, and full fine-tuning. The results suggest that HZO performs competitively or slightly better on several benchmarks. The appendix further provides proofs for subspace optimization and convergence, along with supplementary results.

**Strengths:**

1. The paper targets a realistic and important problem, i.e. fine-tuning large LLMs in low-resource settings.
2. Experiments span multiple models (OPT, LLaMA, RoBERTa) and several benchmark tasks (SST-2, RTE, CB, etc.).
3. The appendix provides empirical guidelines for hyperparameters (r, ν), which could be useful for practitioners.

**Weaknesses:**

1. The method primarily modifies MeZO with low-rank perturbations and FO/ZO inter-layer mixing, without introducing new theoretical principles.
2. Critical details such as environment configuration, metric definitions, and baseline fairness are missing.
3. Explanations are verbose and sometimes unclear, lacking visual support and coherent flow.
4. Conclusions are generalized from narrow experimental cases, and the OPT-175B fine-tuning claim is not empirically supported.

**Questions:**

1. Could the authors provide detailed training time and energy consumption for the OPT-175B experiments, along with reproducible code or scripts?
2. Can Section §3.5 dynamic scheduler be elaborated with a theoretical explanation or diagrammatic illustration, showing overlap mechanisms and communication, i.e. computation balance?
3. Table 4 indicates substantial throughput degradation in FP16 for smaller models, i.e. how do the authors justify this, and is there any mitigation strategy?
4. In the convergence proof, is the assumed low-rank subspace fixed or dynamically updated during training? If fixed, why is that assumption reasonable; if adaptive, how does it affect the validity of the convergence argument?

---

### Official Review · Reviewer_JLbk · 2025-11-01

**Soundness:** 3
**Presentation:** 2
**Contribution:** 2
**Rating:** 2
**Confidence:** 3

**Summary:**

The manuscript presents HZO, a hybrid fine-tuning framework for large language models that combines a low-rank zeroth-order estimator with a statistically stable subspace-sampling schedule, intermittently applies first-order updates to output-proximal layers, and uses a three-stream CPU↔GPU pipeline to hide the latency of dual forward passes.

**Strengths:**

1. The LGE design and sample U every step / sample V every ν steps is a principled way to reduce variance while keeping memory small; Algorithm 1 operationalizes it cleanly.

2. Doing ZO on shallow layers and FO only on output‑proximal layers cuts activation memory yet stabilizes progress, FO only trigger is simple and effective.

**Weaknesses:**

1. Alg. 1 allows either “ZO‑then‑FO” or “FO‑then‑ZO” at resample steps, but the paper does not study the order’s effect;  ν‑ablations are mostly on OPT‑1.3B (Table 10), limiting generality to larger scales.

2.. The convergence proof assumes per‑period V with r I, while Algorithm 1 samples V∼N(0,1) without explicit orthonormalization/normalization.

3. Several claims with no/limited evidence: (1) The abstract says OPT‑175B can be fine‑tuned on “17 GB” GPU memory, while the intro claims a “single 24 GB RTX4090” (2) for OPT‑175B the paper provides only memory/throughput, no accuracy on any task, making the “fine‑tune OPT‑175B” claim empirically incomplete. (3) The intro argues ZO is attractive when activations dominate memory in long context settings, yet there are no >8k/16k context evaluations. (4) it describes pre‑allocation, buckets, and 3 CUDA streams; however there is no ablation with the scheduler disabled, no overlap ratio, and no CPU/NVLink/PCIe specifics—so the engineering contribution cannot be independently assessed.

4. Only compares with MeZO, lack of other ZO baselines.

**Questions:**

The paper notes extra FLOPs due to regenerating U,V from seeds, is there any reports?


duplicate references (last two)

---

### Official Review · Reviewer_Qh5m · 2025-11-02

**Soundness:** 2
**Presentation:** 2
**Contribution:** 1
**Rating:** 2
**Confidence:** 4

**Summary:**

This paper proposes Hybrid Zeroth-Order Optimization (HZO), combining low-rank ZO gradient estimation, inter-layer FO–ZO hybrid optimization, and a dynamic CPU–GPU scheduling strategy for memory-efficient fine-tuning of LLMs. While the method achieves large memory savings and stable convergence, its core components—dynamic scheduling and hybrid optimization—closely resemble existing works ([1], [2]). The novelty is thus limited, and several design claims are insufficiently validated.

[1] ZO2: Scalable Zeroth-Order Fine-Tuning for Extremely Large Language Models with Limited GPU Memory

[2] Revisiting Zeroth-Order Optimization for Memory-Efficient LLM Fine-Tuning: A Benchmark

**Strengths:**

The paper addresses an important problem—memory-efficient ZO fine-tuning for large models—and the implementation details (hybrid scheme, CPU–GPU offloading) are practical and well described.

**Weaknesses:**

Despite its clear motivation, the paper’s novelty is minimal.

1. The proposed dynamic scheduling and memory reuse mechanism closely mirrors ZO2 [1], which already uses CPU–GPU overlap and asynchronous stream scheduling.

2. The inter-layer hybrid FO–ZO optimization essentially replicates the idea explored in [2], offering only minor procedural differences (resample-triggered FO).

3. The low-rank ZO gradient component is conceptually weak: per-layer ZO perturbations and updates are already lightweight, and introducing low-rank sampling adds complexity without clear benefits. Also, if the authors wish to emphasize low-rank adaptation, a direct comparison with LoRA would be necessary.

4. The literature review is shallow, overlooking several recent ZO-based fine-tuning works (e.g., [1], [2]). The related work section needs much deeper contextualization.

5. Experimental coverage is insufficient. OPT models are relatively old; it would be more convincing to test on LLaMA or Qwen series. Moreover, the paper reports only fine-tuning metrics but lacks zero-shot or out-of-domain evaluations, which are essential to assess catastrophic forgetting.

6. Ablation studies isolating the contribution of each proposed component (low-rank estimator, stable sampling, hybrid scheduling) are missing, making it difficult to judge where the real gains come from.

**Questions:**

See "Weaknesses"

---

### Meta-Review · Area_Chair_UqVz · 2026-01-05

**Summary:**

The paper proposes a memory-efficient zeroth-order optimization for fine-tuning large language models. It adopts a low-rank approximation to the ZO gradient and is combined with inter-layer FO–ZO hybrid optimization. It also proposes a dynamic CPU–GPU scheduling strategy. Although the paper studies an important problem, the reviewers share concerns over the limited novelty of applying LORA to ZO without further justification and insufficient experiments to support its claims.

**Reviewer Concerns:**

There is no rebuttal provided by the author. Since there is no rebuttal, I believe the reviewers' concerns still hold with limited novelty and insufficient experiments for the paper's claim.

**Reviewer Scores:**

No since there is no rebuttal.

---

### Decision · Program_Chairs · 2026-01-26

Reject